# ADVERSARIAL TRAINING: EMBEDDING ADVERSARIAL PERTURBATIONS INTO THE PARAMETER SPACE OF A NEURAL NETWORK TO BUILD A ROBUST SYSTEM

## ABSTRACT

Adversarial training, in which a network is trained on both adversarial and clean examples, is one of the most trusted defense methods against adversarial attacks. However, there are three major practical difficulties in implementing and deploying this method - expensive in terms of extra memory and computation costs; accuracy trade-off between clean and adversarial examples; and lack of diversity of adversarial perturbations. Classical adversarial training uses fixed, precomputed perturbations in adversarial examples (input space). In contrast, we introduce dynamic adversarial perturbations into the parameter space of the network, by adding perturbation biases to the fully connected layers of deep convolutional neural network. During training, using only clean images, the perturbation biases are updated in the Fast Gradient Sign Direction to automatically create and store adversarial perturbations by recycling the gradient information computed. The network learns and adjusts itself automatically to these learned adversarial perturbations. Thus, we can achieve adversarial training with negligible cost compared to requiring a training set of adversarial example images. In addition, if combined with classical adversarial training, our perturbation biases can alleviate accuracy trade-off difficulties, and diversify adversarial perturbations.

## 1 INTRODUCTION

Neural networks have lead to a series of breakthroughs in many fields, such as image classification tasks (He et al., 2016), natural language processing (Devlin et al., 2018). Model performance on clean examples was the main evaluation criterion for these applications until the realization of the adversarial example phenomenon by Szegedy et al. (2013); Biggio et al. (2013). Neural networks were shown to be vulnerable to adversarial perturbations: carefully computed small perturbations added to legitimate clean examples (adversarial examples, Fig. 1a) can cause misclassification on state-of-the-art machine learning models. Thus, building a deep learning system that is robust to both adversarial examples and clean examples has emerged as a critical requirement.

Researchers have proposed a number of adversarial defense strategies to increase the robustness of a deep learning system. Adversarial training, in which a network is trained on both adversarial examples ($x_{adv}$) and clean examples ($x_{cln}$) with true class labels $y$, is one of the few defenses against adversarial attacks that withstands strong attacks. Adversarial examples are the summation of adversarial perturbations lying inside the input space ($\delta_I$) and clean examples: $x_{adv} = x_{cln} + \delta_I$ (Fig. 1a). Given a classifier with a classification loss function $L$ and parameters $\theta$, the objective function of adversarial training is:

$$\min_{\theta} L(x_{cln} + \delta_I, x_{cln}, y; \theta) \tag{1}$$

For example, Goodfellow et al. (2014) used adversarial training and reduced the test set error rates from 89.4% to 17.9% on adversarial examples of the MNIST dataset. Huang et al. (2015) built a robust model against adversarial examples by punishing misclassified adversarial examples.

Despite the efficacy of adversarial training in building a robust system, there are three major practical difficulties while implementing and deploying this method. **Difficulty one: adversarial training**

**is expensive in terms of memory and computation costs.** Producing an adversarial example requires multiple gradient computations. In a practical scenario, we further produce more than one adversarial examples for each clean example (Tramèr et al., 2017). We need to at least double the amount of memory, to store those adversarial examples alongside the clean examples. In addition, during adversarial training, the network has to train on both clean and adversarial examples; hence, adversarial training requires at least twice the computation power than just training on clean examples. For example, even on reasonably-sized datasets, such as CIFAR-10 and CIFAR-100, adversarial training can take multiple days on a single GPU. As a consequence, although adversarial training remains among the most trusted defenses, it has only been within reach for research labs having hundreds of GPUs [[[ref]]]. **Difficulty two: accuracy trade-off between clean examples and adversarial examples** - although adversarial training can improve the robustness against adversarial examples, it sometimes hurts accuracy on clean examples. There is an accuracy trade-off between the adversarial examples and clean examples (Di et al., 2018; Raghunathan et al., 2019; Stanforth et al., 2019; Zhang et al., 2019). Because most of the test data in real applications are clean examples, test accuracy on clean examples should be as good as possible. Thus, this accuracy trade-off hinders the practical usefulness of adversarial training because it often ends up lowering performance on clean examples. **Difficulty three: lack of diversity of adversarial perturbations** - even though one might have sufficient computation resources to train a network on both adversarial and clean examples, it is unrealistic and expensive to introduce all unknown attack samples into the adversarial training. For example, Tramèr et al. (2017) proposed Ensemble Adversarial Training which can increase the diversity of adversarial perturbations in a training set by generating adversarial perturbations transferred from other models (they won the competition on Defenses against Adversarial Attacks). Thus, broad diversity of adversarial examples is crucial for adversarial training.

To solve the above three practical difficulties, we dive into details of adversarial training and analyze the causes of these difficulties. **The cause of difficulty one:** In Fig. 1b, a classical deep convolutional network is trained on both clean examples and adversarial examples during adversarial training. Since the adversarial examples are the summation of clean examples and adversarial perturbations (Fig. 1a), the adversarial training uses duplicate information from clean examples and the clean portion of perturbed examples. If we could use information from clean examples only once and generate corresponding adversarial perturbations during the training of clean examples, we could reduce computation costs significantly. **The cause of difficulty two:** to reduce computation costs, one might train the neural network on only adversarial examples. Since adversarial examples are the summation of clean examples and adversarial perturbations, it contains overlapped information from both clean examples and adversarial perturbations. However, failing to incorporate intact information of clean examples hurts test accuracy on clean examples (Di et al., 2018; Raghunathan et al., 2019). **The cause of difficulty three:** even though one might have sufficient computation resources and can afford training on both clean and adversarial examples, the finite amount of adversarial examples still limits the diversity of adversarial perturbations. Thus, the lack of diversity of adversarial perturbations decreases test accuracy on both clean and adversarial examples (Tramèr et al., 2017).

Here, we introduce a new *adversarial perturbation bias* ($\delta_{AP}$) to the last few fully connected layers of the deep convolutional network, replacing the normal bias term (Fig. 1c). The main novelty is that instead of using a fixed, precomputed adversarial perturbations in adversarial examples (input space), we introduce dynamic adversarial perturbations into the parameter space of the network, lying inside the adversarial perturbation bias. During training on clean examples, the adversarial perturbation bias automatically creates and stores adversarial perturbations by recycling the gradient information computed and through updating using the Fast Gradient Sign Method (FGSM) Goodfellow et al. (2014) with the objective function:

$$\min_{\theta, \delta_{AP}} L(x_{cln}, y; \theta, \delta_{AP}) \tag{2}$$

Where $x_{cln}$ is a clean example with true class $y$, $\theta$ is the network parameters, $\delta_{AP}$ is adversarial perturbation bias, and $L$ is the classification loss function. Both adversarial perturbations in adversarial examples and in adversarial perturbation biases are calculated using FGSM. The only difference is that adversarial perturbations in adversarial examples lie inside the input space and adversarial perturbations in adversarial perturbation bias lie inside the parameter space. Thus, the adversarial perturbation biases play a role like adversarial examples in the input space, but they can inject the learned adversarial perturbations directly to the network parameter space. Then, the network learns and adjusts itself automatically to these injected adversarial perturbations. Thus, it is a robust system against adversarial attacks. During training only on clean examples, the network with adversarial

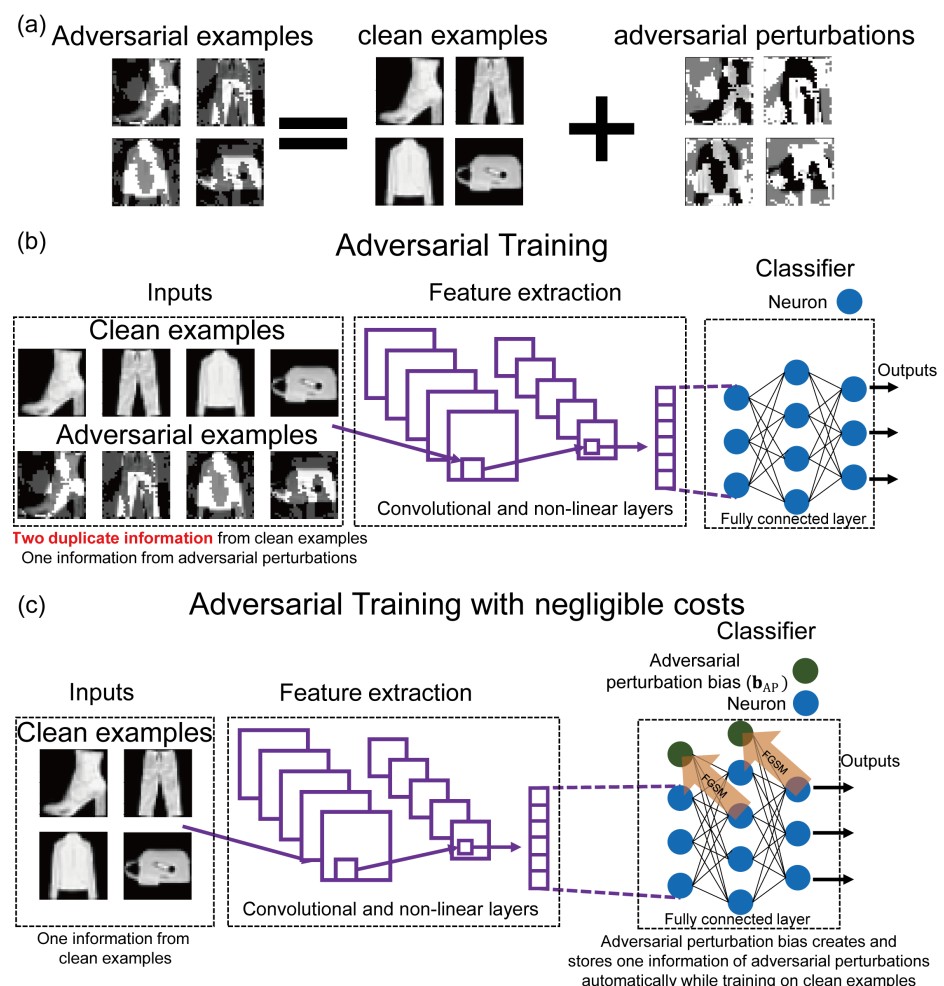

Figure 1: (a) Adversarial examples are a summation of clean examples and adversarial perturbations. (b) Adversarial training for classical deep convolutional network model. Adversarial training on both clean and adversarial examples requires two duplicate information from clean examples and one information on adversarial perturbations. Thus, the adversarial training procedure is expensive. (c) Solving the expensive training procedure. By introducing adversarial perturbation biases to the fully connected layers of a deep convolutional network, we can achieve adversarial training with negligible costs. While training only on clean examples, adversarial perturbation biases automatically create and store adversarial perturbations by recycling the gradient information computed when updating model parameters. The network learns and adjusts itself to these injected adversarial perturbations. Thus, the adversarial perturbation bias increases the robustness against adversarial examples.

perturbation bias shows largely improved test accuracy on adversarial examples, with negligible extra costs. In addition, during classical adversarial training combined with our approach, the network with adversarial perturbation bias can alleviate the accuracy trade-off and diversify available adversarial perturbations. To show the efficacy of the network with adversarial perturbation bias on the above three difficulties, we consider three different scenarios that correspond to different abilities to access increasing computation power.

## 2 RELATED WORK AND BACKGROUND INFORMATION

**Attack models and algorithms:** Most of the attack models and algorithms focus on causing misclassification of target classifiers. They find an adversarial perturbation $\delta_I$. Then, they create an

adversarial example by adding the adversarial perturbation to a clean example $x_{cln}$: $x_{adv} = x_{cln} + \delta_I$. The adversarial perturbation sneaks the clean example $x_{cln}$ out of its natural class and into another. Given a fixed classifier with parameters $\theta$, a clean example $x_{cln}$ with true label y, and a classification loss function $L$, the bounded non-targeted adversarial perturbation $\delta_I$ is computed by solving:

$$\max_{\delta_I} L(x_{cln} + \delta_I, y; \theta) \ , \ subject \ to \ ||\delta_I||_p \leq \epsilon \tag{3}$$

where $||.||_p$ is some $l_p$-norm distance metric, and $\epsilon$ is the adversarial perturbation budget.

In this work, we consider the most popular non-targeted method - Fast Gradient Sign Method (FGSM) by Goodfellow et al. (2014), in the context of $l_\infty$-bounded attacks: $x_{adv} = x_{cln} + \epsilon \ sign(\nabla_{x_{cln}} L(x_{cln}, y, \theta))$

**White box attack model:** In white box attacks (Qiu et al., 2019), the adversaries have complete knowledge about the target model, including algorithm, data distribution and model parameters. The adversaries can generate adversarial perturbations by identifying the most vulnerable feature space of the target model. In this work, we use white box Fast Gradient Sign Method with $\epsilon = 0.3$ to generate adversarial perturbations.

**Modification of bias terms:** Modifying the structure of bias terms in the fully connected layer can be beneficial. Wen & Itti (2019) used bias units to store the beneficial perturbations (opposite to the well-known adversarial perturbations). Wen & Itti (2019) showed that the beneficial perturbations, stored inside task-dependent bias units, can bias the network outputs toward the correct classification region for each task, allowing a single neural network to have multiple input to output mappings. Multiple input to output mappings alleviate the catastrophic forgetting problem (McCloskey & Cohen, 1989) in sequential learning scenarios (a previously learned mapping of an old task is erased during learning of a new mapping for a new task). Here, we leverage a similar idea. During training on clean examples, adversarial perturbation bias automatically creates and stores the adversarial perturbations. The network automatically learns how to adjust to these adversarial perturbations. Thus, it helps us to build a robust model against adversarial examples.

## 3 NETWORK STRUCTURE AND ALGORITHM FOR A NEURAL NETWORK WITH ADVERSARIAL PERTURBATION BIAS.

**Two different structures for adversarial perturbation bias:** naive adversarial perturbation bias and multimodal adversarial perturbation bias. Naive adversarial perturbation bias ($\delta_{AP}$) is just a normal bias term of the fully connected layer. The only difference is that during backpropagation, we update the naive adversarial perturbation bias using the Fast Gradient Sign Method. For multimodal adversarial perturbation bias, we design the multimodal adversarial perturbation bias ($\delta_{AP} = m_{AP}W_{AP}$) as a product of bias memories ($m_{AP} \in R^{1 \times h}$) and bias weight ($W_{AP} \in R^{h \times n}$), where $n$ is the number of neurons in the fully connected layer, and $h$ the number of bias memories. Multimodal adversarial perturbation bias has more degrees of freedom to better fit a multimodal distribution. Thus it could yield better results than the naive adversarial perturbation bias. The update rules for the network with naive (blue) and multimodal (red) adversarial perturbation bias are shown in Alg. 1 and Alg. 2.

---

**Algorithm 1** Forward rules for both naive and multimodal adversarial perturbation bias

> **Input:** $X_{activations}$— Activations from the last layer
> $\boldsymbol{\delta_{AP}}$— Adversarial perturbation bias
> **Output:** $\boldsymbol{Y = W X_{activations} + \delta_{AP}},$      where: $\boldsymbol{W}$— Normal neuron weights.

---

## 4 THE FUNCTION OF ADVERSARIAL PERTURBATION BIAS IN THREE DIFFERENT SCENARIOS

**Scenario 1, adversarial training with negligible costs:** a company with modest computation resources can only afford training the network on clean examples, but still wants to have a moderately robust system against adversarial examples. During training on clean examples, when updating model

---

**Algorithm 2** Backward rules for naive and multimodal adversarial perturbation bias

---

**Notations:** $W$ — Normal neuron weights    $X_{activations}$ — Activations from the last layer
$\delta_{AP}$ — Adversarial perturbation bias (native or multimodal)
$\mathbf{m_{AP}}$ — Bias memories for multimodal adversarial perturbation bias
$\mathbf{W_{AP}}$ — Bias weights for multimodal adversarial perturbation bias
$\epsilon$ — Adversarial perturbation budgets for Fast Gradient Sign Method

During the training:
  **Input:**  **Grad** — Gradients from the next layer
  **output:** $\mathbf{dW} = \mathbf{Grad}.dot(\mathbf{dX}_{activations}^T)$ // gradients for the normal neuron weights
      $\mathbf{dX}_{activations} = \mathbf{W^T}.dot(\mathbf{Grad})$      // gradients for activations to last layer
      For naive adversarial perturbation bias:
        $\boldsymbol{d\delta_{AP}} = \epsilon sign(\sum_{n=1}^{number\ of\ samples} \mathbf{Grad})$
              // gradients for the naive adversarial perturbation bias using FGSM

      For multimodal adversarial perturbation bias:
        $\mathbf{dm_{AP}} = \epsilon\ sign\ (\mathbf{W_{AP}^T}.dot(\mathbf{Grad}))$ // gradients for the bias memories of
                      multimodal adversarial perturbation bias using FGSM
        $\mathbf{dW_{AP}} = \mathbf{Grad}.dot(\mathbf{m_{AP}^T})$
            // gradients for the bias weights of multimodal adversarial perturbation bias

After the training:
      For multimodal adversarial perturbation bias:
        Keep the $\boldsymbol{\delta_{AP}}$ as the product of $\mathbf{m_{AP}}$ and $\mathbf{W_{AP}}$
        Delete $\mathbf{m_{AP}}$ and $\mathbf{W_{AP}}$ to reduce memory and parameter costs

---

parameters with objective function Eqn. 2, in the backward pass of backpropagation, adversarial perturbation bias automatically creates and stores adversarial perturbations (calculated from clean examples). In the forward pass, the adversarial perturbation bias injects the learned adversarial perturbations to the network. As a result, the neural network learns the structure of these adversarial perturbations and adjusts itself to against these injected adversarial perturbations automatically during the training on clean examples. Although the network cannot be trained on the adversarial examples to access the information of adversarial perturbations directly because of the modest computation power, adversarial perturbation bias can still provide the information of adversarial perturbations to the network. Thus, adversarial perturbation bias can improve the model's robustness against adversarial examples (see Results) and maintain the highest test accuracy on clean examples while barely increase any computation costs. For naive adversarial perturbation bias, we do not introduce any extra computation costs beyond FGSM. For multimodal adversarial perturbation, the extra computation costs are further increased by a matrix multiplication in the forward pass and two dots products in the backward pass per fully connected layer. The state-of-the-art deep convolutional structure might have up to three fully connected layers as a classifier, yielding up to three extra multiplication and six extra dot products computation costs. The extra computation costs are negligible comparing to generating a huge amount of extra adversarial examples and training the neural network on both clean and adversarial examples.

**Scenario 2, counteracting the adversarial perturbations lying inside of the adversarial examples:** to have strong robustness against adversarial examples, a company with moderate computation resources can only afford to generate adversarial examples and train a network only on adversarial examples. Training only on adversarial examples can achieve high robustness against adversarial examples, but it decreases the test accuracy on the clean examples. During training on adversarial examples, when updating model parameters, in the backward pass of backpropagation, adversarial perturbation bias automatically creates and stores adversarial perturbations (calculated from the adversarial examples) by recycling the gradient information computed. In the forward pass, the adversarial perturbation bias injects the learned adversarial perturbations to the network. There is a high chance that the adversarial perturbations ($\delta_{AP}$ calculated from adversarial examples) lying inside the parameter space of neural network (adversarial perturbation bias) would counteract the adversarial perturbations ($\delta_I$ calculated from clean examples) lying inside the input space (images of adversarial

examples). In mathematical term, in the view of the network, $x_{adv} + \delta_{AP} = x_{cln} + \delta_I + \delta_{AP} \approx x_{cln}$, where $\delta_I$ counteract with the $\delta_{AP}$. Adversarial perturbation bias can convert some adversarial examples into clean examples because of the counteraction. Thus, it largely improves the testing accuracy on the clean examples (see Results), while it still maintains a high testing accuracy against adversarial examples.

**Scenario 3, diversify the adversarial perturbations:** a company with abundant computation resources can afford to train a network on both clean and adversarial examples. In the generation of adversarial examples, the adversarial perturbations lying inside the adversarial examples are calculated from the clean examples. In comparison, during the training of the network on both clean and adversarial examples, in the backward pass, the adversarial perturbation bias creates and stores adversarial perturbations calculated from both clean and adversarial images by recycling the gradient information computed. This allows the adversarial perturbation bias to diversify the available adversarial perturbations and to inject these varied adversarial perturbations to the network. In addition, adversarial perturbations lying inside the adversarial examples are fixed and precomputed. In addition, the adversarial perturbations lying inside the adversarial perturbation bias change dynamically for every training epoch. This further increases the variations of adversarial perturbations during the training. In mathematical representation, the diversified group has: clean examples $x_{cln}$, adversarial examples $x_{adv}$, perturbed clean examples $x_{cln} + \delta_{AP}$ and perturbed adversarial examples $x_{adv} + \delta_{AP}$. Adversarial perturbation bias can diversify the adversarial perturbations available to the neural network using adversarial training. As a result, the diversity improves the testing accuracy on both clean and adversarial examples (see Results).

## 5 EXPERIMENTS

### 5.1 DATASET AND NETWORK STRUCTURE

MNIST (LeCun et al., 1998) is a dataset with handwritten digits, has a training set of 60,000 examples, and a test set of 10,000 examples. FashionMnist (Xiao et al., 2017) is a dataset with article images, has a training set of 60,000 examples, and a test set of 10,000 examples. We use a LeNet (LeCun et al., 1998) as a classifier (classical LeNet). Then, we create our version of LeNet (LeNet with adversarial perturbation bias) by adding adversarial perturbation bias into the fully connected layers, replacing the normal bias. We generate the adversarial examples using Fast Gradient Sign Method for both classical LeNet and LeNet with adversarial perturbation bias.

### 5.2 QUANTITATIVE RESULTS

We test the LeNet with adversarial perturbation bias and classical LeNet on both clean and adversarial examples from MNIST and FashionMNIST dataset, under 3 different computation budgets: Negligible costs (Training only on clean examples), moderate extra costs (Training only on adversarial examples) and high extra costs (Training on both clean and adversarial examples).

**Negligible cost - training only on clean examples: our method can largely increase test accuracy on adversarial examples and slightly increase test accuracy on clean examples.** When the neural network can only be trained on clean examples because of modest computation power, LeNet with adversarial perturbation bias achieves a slightly higher test accuracy on clean examples than classical Lenet (Fig. 2B MNIST: 99.17% vs. 99.01%, FashionMNIST: 89.54% vs. 89.17%). In addition, for the test accuracy on adversarial examples (Fig. 2A), classical Lenet can only achieve 18.08% on the MNIST dataset and 11.87% on the FashionMNIST dataset. In comparison, LeNet with adversarial perturbation bias can achieve 98.88% on the MNIST dataset and 53.88% on the FashionMNIST dataset. Thus, for companies with modest computation resources, adversarial perturbation bias can help a system achieve moderate robustness against adversarial examples, while only introducing negligible computation costs (e.g., on FashionMNIST, our method only uses 59% training time compared to the training time of adversarial training with just one adversarial example per clean example, saving 43.51 minutes training time for 500 training epochs on NVIDIA Tesla-V100 platform. The saving would be huge on a larger dataset such as Imagenet (Deng et al., 2009)).

**Moderate extra costs - training only on adversarial examples: our method can slightly increase the test accuracy on adversarial examples and largely increase the test accuracy on clean examples.** In the generation of adversarial examples, it takes multiple gradient computation through

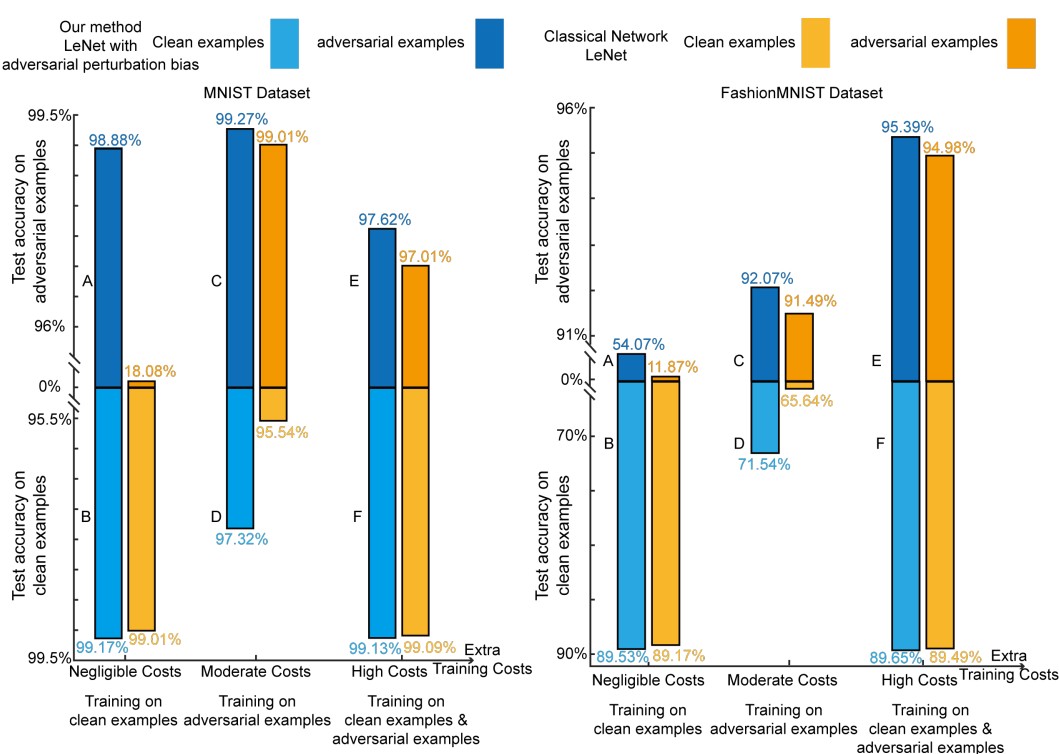

Figure 2: Performance of LeNet with adversarial perturbation bias (our method) versus classical network LeNet under three different computation power budgets for three different scenarios. The adversarial examples are generated using FGSM ($\epsilon = 0.3$). 1. Negligible costs scenario (A, B): training neural network only on clean examples. LeNet with adversarial perturbation bias demonstrates much increased test accuracy on adversarial examples, and slightly increased test accuracy on clean examples compared to classical LeNet. 2. Moderate extra costs scenario (C, D): training neural network only on adversarial examples. LeNet with adversarial perturbation bias demonstrates largely increased test accuracy on clean examples and slightly increased test accuracy on adversarial examples compared to classical LeNet. 3. High extra costs scenario (E, F): training the neural network on both clean and adversarial examples. LeNet with adversarial perturbation bias demonstrates slightly increased test accuracy on both clean and adversarial examples compared to classical LeNet.

backpropagation to calculate adversarial perturbations. As a consequence, training only on adversarial examples is slightly more expensive than training only on clean examples. On classical networks, although training only on adversarial examples can achieve a high test accuracy on adversarial examples (Fig. 2C classical LeNet: MNIST 99.01%, FashionMNIST 91.49%), it hurts the test accuracy on clean examples. For example, for classical LeNet, training on clean examples can achieve test accuracy - 99.01% for the MNIST dataset and 89.17% for the FashionMNIST dataset (Fig. 2B). For classical LeNet, training on the adversarial examples can only achieve a much worse testing accuracy - 95.54% for MNIST dataset and 65.64% for FashionMNIST dataset (Fig. 2D). In comparison, for LeNet with adversarial perturbation bias, training on the adversarial examples can achieve a testing accuracy - 97.72% for MNIST dataset and 71.54% for FashionMNIST dataset (Fig. 2D). This accuracy is still worse than the accuracy of training only on clean examples, but it is much better than the accuracy of classical LeNet training only on adversarial examples. Thus, for companies with moderate computation resources, we build a strong robust system against adversarial examples while achieving a better testing accuracy on clean examples.

**High extra costs - training on both clean and adversarial examples: our method can diversify the adversarial perturbations, so it can slightly increase the test accuracy on both clean and adversarial examples.** LeNet with adversarial perturbation bias can achieve slightly higher accuracy

on clean examples than classical Lenet (Fig. 2F MNIST 99.13% vs. 99.09%, FashionMNIST 89.65% vs. 89.49%). In addition, LeNet with adversarial perturbation bias can achieve slightly higher accuracy on adversarial examples than classical LeNet (Fig. 2E MNIST 97.62% vs. 97.01%, FashionMNIST 95.39% vs. 94.98%). Even for companies with abundant computation resources, it is still helpful to adapt our adversarial perturbation bias because it diversifies the adversarial perturbations.

### 5.3 INFLUENCE OF THE ADVERSARIAL PERTURBATION BUDGETS AND STRUCTURE OF ADVERSARIAL PERTURBATION BIAS

**Adversarial perturbation budgets:** The higher the adversarial perturbation budgets, the higher the chance it can successfully attack a neural network. However, attacks with higher adversarial perturbation budgets are easier to detect by a program or by humans. For example, $\epsilon = 0.3$ (Fig. 1a) represents very high noise, which makes FashionMNIST images difficult to classify, even by humans. But the distribution differences between the adversarial examples and clean examples are so large that they can be easily captured by defense programs. Thus, $\epsilon \leq 0.15$ is a good attack since the differences caused by adversarial perturbations are too small to be detected by most defense programs. For small adversarial perturbations (Fig. 3a $\epsilon \leq 0.15$), by just training on clean images, LeNet with adversarial perturbation bias achieves moderate robustness against adversarial examples with negligible costs. Thus, it is really beneficial to adapt our method for companies with modest computation power, who still want to achieve moderate robustness against adversarial examples. **Structure of adversarial perturbation bias:** under the FGSM attacks ($\epsilon = 0.3$, Fig. 3b), multimodal adversarial perturbation bias works better than naive perturbation bias on MNIST. However, naive adversarial perturbation bias works better than multimodal perturbation bias on FashionMNIST. Thus, the structure of adversarial perturbation bias is a hyperparameter for different datasets. The number of bias memories of multimodal adversarial perturbation bias is another hyperparameter. If we have insufficient bias memories, there is not enough degrees of freedom to learn a good multimodal distribution. If we have excessive bias memories, the model overfits the multimodal distribution.

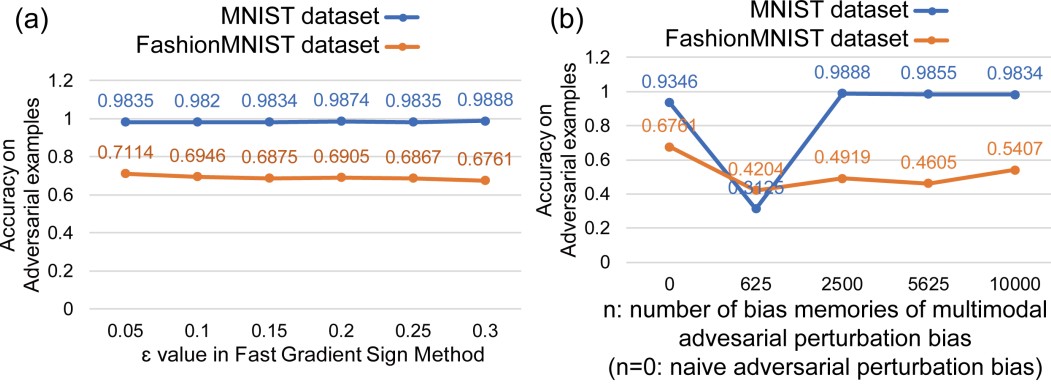

Figure 3: Test accuracy on adversarial examples after training only on clean examples from MNIST (Blue) or FashionMNIST (Orange) datasets. (a): Influence of different adversarial perturbation budgets ($\epsilon$) on LeNet with adversarial perturbation bias. (b): Influence of different structures of adversarial perturbation bias on LeNet under adversarial attack: FGSM $\epsilon = 0.3$.

## 6 CONCLUSION

In this paper, we propose a new method to solve the three major practical difficulties while implementing and deploying adversarial training, by embedding adversarial perturbation into the parameter space (adversarial perturbation bias) of neural network. There are three major contributions that benefit for companies with different levels of computation resources - 1. Modest Computation power: adversarial training with negligible costs. 2. Moderate computation power: alleviate the accuracy trade-off between clean examples and adversarial examples. 3. Abundant computation power: diversify the adversarial perturbations available to the adversarial training.

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
