# OpenReview forum: "Adversarial Training: embedding adversarial perturbations into the parameter space of a neural network to build a robust system"
_ICLR.cc/2020/Conference — Reject_

### Official Review · AnonReviewer3 · 2019-10-16
**Official Blind Review #3**

**Rating:** 1

**Review:**


This paper proposes to introduce adversarial perturbations into intermediate layers of a neural network, to achieve more efficient adversarial training.
The idea has been proposed before. The paper is based on, and perpetrates, a number of fundamental misconceptions about adversarial examples. A lack of many relevant citations indicates that the authors are not familiar with the related work done in this field over the past three years. This paper is currently clearly below the bar for ICLR.

We recommend that the authors read, understand, and cite at least the list of papers below before revising their paper:

- Madry et al., "Towards deep learning models resistant to adversarial attacks", 2017
- Sabour et al., "Adversarial Manipulation of Deep Representations", 2016
- Tsipras et al., "Robustness May Be at Odds with Accuracy", 2019
- Carlini & Wagner., "Towards Evaluating the Robustness of Neural Networks", 2017
- Carlini et al., "On evaluating adversarial robustness", 2019

In particular, this should help clarify the following misconceptions:
- Adversarial training does not require training on both clean and adversarial examples. The best results are often obtained by only training on adversarial examples.
- Adversarial training does not require extra memory
- While expensive, adversarial training does not require days of computation on hundreds of GPUs (not even on ImageNet)
- There is no evidence that adversarial training with PGD produces "non-diverse" adversarial examples (for the chosen perturbation set)
- FGSM is not a good benchmark for training or evaluation

**Experience Assessment:**

I have published in this field for several years.

**Review Assessment: Checking Correctness Of Derivations And Theory:**

N/A

**Review Assessment: Checking Correctness Of Experiments:**

N/A

**Review Assessment: Thoroughness In Paper Reading:**

I read the paper thoroughly.

---

### Official Review · AnonReviewer2 · 2019-10-23
**Official Blind Review #2**

**Rating:** 3

**Review:**

This paper proposes perturbation biases as a counter-measure against adversarial perturbations. The perturbation biases are additional bias terms that are trained by a variant of gradient ascent. The method imposes less computational costs compared to most adversarial training algorithms. In their experimental evaluation, the algorithm achieved higher accuracy on both clean and adversarial examples.

This paper should be rejected because the proposed method is not well justified either by theory or practice. Experiments are weak and do not support the effectiveness of the proposed method.

Major comments:
Since the evaluations of defense algorithms are often misleading [1], it requires throughout experiments or theoretical certifications to confirm the effectiveness of defense methods. However, the experiment configuration in this paper is not satisfactory to demonstrate the robustness of defended networks. The followings are a list of concerns.
1) Experiments are limited to small datasets and networks. Since some phenomena only appear in larger datasets [2], there is a concern that the proposed method also works on other datasets.
2) The attack algorithm used for the evaluation is weak. We can confirm this by observing the label leakage [2] in the experimental results. It is hard to judge which defenses are most effective, even within the tested datasets and models.
3) The "adversarial training" baseline used in the experiment is weird. Adversarial training typically generates adversarial examples during the process of the neural networks' optimization instead of using precomputed adversarial examples. Baseline methods should be stronger, for example, adversarial training with PGD [3].

[1] Athalye et al. "Obfuscated Gradients Give a False Sense of Security: Circumventing Defenses to Adversarial Examples." ICML 2018
[2] Kurakin et al. "ADVERSARIAL MACHINE LEARNING AT SCALE." ICLR 2017
[3] Madry et al. "Towards Deep Learning Models Resistant to Adversarial Attacks." ICLR 2018

**Experience Assessment:**

I have published one or two papers in this area.

**Review Assessment: Checking Correctness Of Derivations And Theory:**

I did not assess the derivations or theory.

**Review Assessment: Checking Correctness Of Experiments:**

I assessed the sensibility of the experiments.

**Review Assessment: Thoroughness In Paper Reading:**

I read the paper at least twice and used my best judgement in assessing the paper.

---

### Official Review · AnonReviewer1 · 2019-10-23
**Official Blind Review #1**

**Rating:** 3

**Review:**

The authors try to tackle the problem of adversarial examples by introducing a special set of bias weights into the neural network. There are serious clarity issues with the writing of this paper. It cites Wen & Itti 2019 but provides few motivations and explanations of algorithmic choices.

My questions include:
  - Is there a separate bias term for clean examples? When is the adversarial bias used? Is it only for adversarial examples?
  - How is the adversarial bias term updated? In Algorithm 2, its gradient is a sum over samples, but over what samples? A mini-batch?
  - What are the multi-modal weights? There is no forward equation on how they are used. We only have their updates in Algorithm 2.

These are all very confusing, given the central importance of Algorithm 2 to the paper. The authors should start with how the new bias terms affect inference and predictions, before going to their updates.

There are also limitations in the experiments. Only Fast Gradient Sign Method (FGSM) is used to generate adversarial examples. The more powerful projected gradient descent should be used for a better test against adversarial examples. Also, only MNIST and FashionMNIST are tested. The authors should consider including CIFAR10, CIFAR100, or other datasets.

Overall I think the issues with clarity and experiments in the paper make it not ready for publication yet.




**Experience Assessment:**

I have published one or two papers in this area.

**Review Assessment: Checking Correctness Of Derivations And Theory:**

I assessed the sensibility of the derivations and theory.

**Review Assessment: Checking Correctness Of Experiments:**

I assessed the sensibility of the experiments.

**Review Assessment: Thoroughness In Paper Reading:**

I read the paper thoroughly.

---

### Decision · Program_Chairs · 2019-12-19

**Decision:**

Reject

**Comment:**

This paper proposes to introduce perturbation biases as a counter-measure against adversarial perturbations. The perturbation biases are additional bias terms that are trained by a variant of gradient ascent. Serious issues were raised in the comments. No rebuttal was provided.